# Using ELP Repeats as a Scaffold for De Novo Construction of Gadolinium-Binding Domains within Multifunctional Recombinant Proteins for Targeted Delivery of Gadolinium to Tumour Cells

**DOI:** 10.3390/ijms23063297

**Published:** 2022-03-18

**Authors:** Natalia V. Pozdniakova, Oxana V. Ryabaya, Alevtina S. Semkina, Vsevolod A. Skribitsky, Alexei B. Shevelev

**Affiliations:** 1N.N. Blokhin National Medical Research Center of Oncology of the Ministry of Public Health of the Russian Federation (N.N. Blokhin NMRCO), Kashirskoe Shosse, 23, 115478 Moscow, Russia; oxa2601@yandex.ru (O.V.R.); skvseva@yandex.ru (V.A.S.); 2Department of Medical Nanobiotechnologies, Pirogov Russian National Research Medical University, Ostrovityanova, 1, 117997 Moscow, Russia; alevtina.semkina@gmail.com; 3Department of Basic and Applied Neurobiology, Serbsky National Medical Research Center for Psychiatry and Narcology, Kropotkinskiy, 23, 119991 Moscow, Russia; 4Institute of Engineering Physics for Biomedicine (PhysBio), National Research Nuclear University MEPhI, Kashirskoe shosse, 31, 115409 Moscow, Russia; 5N.I. Vavilov Institute of General Genetics RAS, Gubkina 3, GSP-1, 119991 Moscow, Russia; shevel_a@hotmail.com

**Keywords:** recombinant protein engineering, ELP, binary radiotherapy, gadolinium contrast agent, F3 peptide, RGD

## Abstract

Three artificial proteins that bind the gadolinium ion (Gd^3+^) with tumour-specific ligands were de novo engineered and tested as candidate drugs for binary radiotherapy (BRT) and contrast agents for magnetic resonance imaging (MRI). Gd^3+^-binding modules were derived from calmodulin. They were joined with elastin-like polypeptide (ELP) repeats from human elastin to form the four-centre Gd^3+^-binding domain (4MBS-domain) that further was combined with F3 peptide (a ligand of nucleolin, a tumour marker) to form the F3-W4 block. The F3-W4 block was taken alone (E2-13W4 protein), as two repeats (E1-W8) and as three repeats (E1-W12). Each protein was supplemented with three copies of the RGD motif (a ligand of integrin αvβ3) and green fluorescent protein (GFP). In contrast to Magnevist (a Gd-containing contrast agent), the proteins exhibited three to four times higher accumulation in U87MG glioma and A375 melanoma cell lines than in normal fibroblasts. The proteins remained for >24 h in tumours induced by Ca755 adenocarcinoma in C57BL/6 mice. They exhibited stability towards blood proteases and only accumulated in the liver and kidney. The technological advantages of using the engineered proteins as a basis for developing efficient and non-toxic agents for early diagnosis of tumours by MRI as well as part of BRT were demonstrated.

## 1. Introduction

The search for new, safe methods of cancer diagnosis and treatment is still relevant. Binary radiotherapy (BRT) is one modern type of radiation therapy, in which two factors are used simultaneously to increase the local dose load on the tumour: (1) direct physical targeting of the radiation beam to the tumour, which is provided by technical means, and (2) administering specific radioenhancers that selectively accumulate in the tumour and enhance the biological effect of radiation. The radioenhancers are usually complex compounds containing high-Z elements (Z ≥ 53, for example gold [Au, Z = 79], bismuth [Bi, Z = 83], hafnium [Hf, Z = 72], and gadolinium [Gd, Z = 64]). The interaction between radiation and high-Z elements triggers a cascade of intracellular reactions that irreversibly damage cancer cells. This ensures a local dose increase during radiation therapy while preserving the surrounding healthy tissues. Gd is attractive among high-Z elements because it also has an abnormally high neutron capture cross-section (2.55 × 10^5^ barns) [1]; that is, it is one of the most effective components of neutron capture therapy (one of the variants of BRT). Besides, Gd^3+^ can act also as positive contrast agent for magnetic resonance imaging (MRI), thereby combining both diagnostic and therapeutic potential. Due to this potential, Gd is a unique element for biomedical applications, in particular, for radiation therapy under the control of MRI, a new method being developed thanks to recent developments in magnetic resonance imaging systems combined with a linear accelerator [2,3].

In this regard, there is increased research into investigating existing and designing new Gd-containing, dose-enhancing agents with improved characteristics, which are first of all capable of selectively accumulating in the tumour and its microenvironment. The question arises whether it is possible to use already existing Gd-containing contrast agents (e.g., Magnevist^®^ [gadopentetic acid], Gadovist, Optimark and Omniscan) with excellent contrast properties for this purpose. There are data demonstrating that low-molecular-weight complex compounds, like Magnevist^®^, despite good results in vitro [4], are often inefficient in clinical trials [5] mainly due to rapid clearance from the bloodstream, lack of specific tropism to tumour tissue, and a tendency to accumulate and be retained in some types of healthy tissues (e.g., in the brain and bones) [6]. More promising results in terms of selectivity (tumourotropy) are shown by Gd-containing nanoparticles. They accumulate in tumours due to the enhanced permeability and retention (EPR) effect [1] caused by the abnormal structure of blood vessels in tumours compared with normal tissues and the presence of numerous tumour-associated macrophages actively capturing insoluble particles. In this research field, only AGuIX^®^ nanoparticles (NH TherAguix, France) have successfully passed phase I clinical trials (NCT02820454, ClinicalTrials.gov) [7] and will be tested in a phase II clinical trial (NCT03818386, ClinicalTrials.gov). AGuIX^®^ nanoparticles with a size of 5 nm have a polysiloxane core surrounded by Gd chelating complexes (usually 10); they were developed as a radiosensitiser for radiation therapy under the control of MRI [8]. However, one should take into account the fact that in recent years the real significance of the EPR effect for nanomedicine in relation to humans has been discussed actively and even disputed [9]. Therefore, the need for new Gd-containing preparations with improved pharmacokinetic characteristics, selective accumulation, sufficient circulation time in the bloodstream, and reduced toxicity, among other factors, remains acute. These new compounds could be generated by functionalising the surface of nanoparticles with specific peptides [10], polysaccharides [11], and other molecules involved in the receptor-mediated accumulation of radiosensitising substances in the tumour itself or surrounding tissues [12,13]. Research is also underway to create new targeted low molecular weight chelating compounds [12].

Recombinant proteins such as a Gd carrier platform provide a promising alternative to synthetic polymers or low-molecular-weight molecules and have several advantages. (1) They are completely biocompatible and biodegradable. (2) Unlike most chemical polymers, they have a strictly deterministic size of the globule and, therefore, their bio-distribution is more uniform. (3) A relatively small number of amino acids is sufficient to create an almost unlimited number of protein molecules with a wide range of physicochemical properties and functions. (4) The production of proteins occurs in biological systems; this eliminates the need for the use of toxic chemicals for the synthesis of a medicinal substance. (5) Finally, because all living beings use proteins, there is a huge natural ‘database’ that can be used as a resource for creating new biomolecules [14,15].

There are a number of works on the design of protein contrast preparations with Gd aimed at (1) enhancing contrast properties, (2) increasing ion selectivity and stability of the carrier complex with gadolinium ion (Gd^3+^), and (3) improving targeting. One group [16] developed a proteinaceous MRI contrast agent (ProCA32) based on rat and human α-parvalbumin to detect micrometastatic liver tumours at an early stage. In another study [17], the protein contrast agent ProCA1.GRP contained a fused gastrin-releasing peptide (GRP) for active targeting and imaging of prostate cancer and showed good retention during intra-tumour administration. In another study, researchers developed a ProCA32.collagen1 block with two Gd^3+^ binding sites and a GGGKKWHCYTYFPHHYCVYG peptide providing affinity for collagen I for visualisation of liver fibrosis [18]. Another group examined a two-domain fused Zarvin protein, which on the one hand contained two high-affinity Gd^3+^ binding sites, and on the other hand was able to bind to the constant part of therapeutic humanised immunoglobulin G (IgG) class antibodies [19]. In the above examples, the number of Gd^3+^ ions per protein molecule does not exceed two, and the active targeting is directed to a single tumour surface marker. Besides, artificial tumour-specific proteins conjugates (compositions) including different types of the listed molecules (proteins, peptides, polymers, and chelators) could be used for the same purpose [20]. This approach provides more accurate targeting, however, multi-stage synthesis complicates the technology of obtaining the medicine and makes its quality standardisation difficult. Taken together, these issues negatively affect the affordability of the drug for the patient and decrease the reproducibility of the results of diagnosis or treatment obtained with its use.

Considering the abovementioned points, this study pursued development of a complex technical approach to create recombinant Gd^3+^-binding proteins that would meet the principal requirements for application in BRT and MRI: (1) Proteins should be able to bind as many Gd ions as possible and hold them firmly enough under physiological conditions; (2) protein complexes with Gd should have contrast characteristics that are not inferior to contrast agents currently used in MRI practice; (3) proteins should be synthesised in the bacterial expression system in a soluble form with a high yield to ensure the technological acceptability of the purification procedure; (4) high in vivo affinity and specificity of the proteins to the tumours should be ensured by independently targeting two tumour markers; and (5) the ligand peptides should be multimerised to achieve a cooperative effect upon targeting the tumour receptor. It is known that oligo- or multimerisation of functional blocks can be used to improve or create de novo certain properties of artificially designed proteins. For example, multimerisation of RGD-containing peptides could lead to a multi-fold increase in the affinity of binding to target cells [21,22], and dimerisation of cell-penetrating peptides could lead to an increase in the efficiency of capture and cytoplasmic localisation [23,24]. However, it is obvious that the multimerisation of individual parts of the protein without taking into account changes in the basic physical and chemical properties of the protein in most cases would lead to adverse consequences, such as instability of the globular structure and low solubility, aggregation, nonspecific binding to various structures in vivo, loss of proteolytic stability, a drop in the level of expression, etc. Therefore, in this work, an algorithm was used for coordinated multiplication of several initially balanced protein functional blocks in terms of physicochemical properties, whereafter the properties of the molecule as a whole did not change or changed insignificantly. In general, artificial proteins were designed by taking into account the fact that they should be immunologically compatible with the human internal environment, non-toxic, and stable during expression in bacteria as well as at all stages of purification and preparation of a complex with metal and upon administration in vivo. To comply with these requirements, first, peptides from the composition of proteins encoded by the human genome were used as initial modules, and second, their functional (biological) and physicochemical (isoelectric point, hydrophobicity profile) properties were considered when choosing the parental blocks.

Such a multilateral task was solved by designing individual functional protein blocks with their subsequent combination and multimerisation in compliance with the abovementioned conditions. First, a Gd-binding domain (4MBS-domain) was designed as the main core, which contained four Gd^3+^-binding centres derived from human calmodulin alternating with elastin-like polypeptide (ELP) repeats derived from human elastin. The latter were used as a structural component that creates a moderately hydrophobic environment and stabilises the overall globular structure of the 4MBS-domain. It was assumed that such moderate hydrophobicity, on the one hand, would give sufficient stability to the Gd^3+^ complex with the chelator peptide and, on the other hand, would not interfere with the access of water molecules to the first coordination sphere of Gd^3+^, which is important for the contrast characteristics of the designed protein. ELP peptides are well known in structural biology; they are non-toxic, non-immunogenic, and have the ability to undergo a controlled phase transition [25,26,27]. Two different tumour-homing motifs were also included in the composition of each protein. One of them, the RGD peptide, has affinity for integrin αvβ3 [28,29,30,31], which isone of the historically first known tumour markers used in anti-cancer therapy. Integrin αvβ3 is abundantly present on the cell surface of solid tumours of almost all known types, as well as on the vascular network of tumours. The RGD peptide was included in three copies in all the proteins developed in this work. As a second tumour-specific ligand, F3 peptide was used, which has affinity for nucleolin [32,33,34,35]. Nucleolin is involved in the development of some pathological conditions, including oncogenesis and viral infection [36]. Increased expression of nucleolin is observed in some solid cancers and blood tumours. Of note, when overexpressed, nucleolin can be localised not only in the nucleolus, but also transported to the plasma membrane [37]. Increased nucleolin in the membrane is observed not only on the surface of some types of tumours, but also on activated lymphocytes; angiogenic endothelial cells; and other cells involved in the processes of inflammation, angiogenesis, lymphangiogenesis, and oncogenesis [38]. Nucleolin has been suggested as a promising target in the development of targeted antitumor drugs [39]. In addition to serving as a tumour-homing component, F3 peptide, which has pronounced cationic properties (pI = 10.77), was used to compensate for the negative charge of the 4MBS block (pI = 4.28) when designing the W4 domain. For this reason, when designing the proteins, the W4 domain including F3 peptide and the 4MBS block underwent multimerisation as a whole. Consequently, three similar proteins with the total number of Gd^3+^-binding sites equal to 4, 8, and 12 were designed by dimerisation and trimerisation of the W4 domain (one, two, and three copies, respectively). Green fluorescent protein (GFP) was introduced to all three artificial proteins as an arbitrary reporter facilitating experiments for testing its biological distribution in different systems.

A special technology for producing genetically stable plasmid constructs with multiple alternating tandem repeats was established. It includes a restriction/ligation step using partially randomised initial blocks containing alternative synonymous codons and interchangeable amino acids in all available positions. Translation initiation of the commercially available vector pRSET-EmGFP (Thermo Fisher Scientific, Waltham, MA, USA) was modified to increased protein yield in *Escherichia coli*. Taking into account the high affinity of MBS for metal ions, the scheme includes a nickel ion (Ni^2+^) purification step after conventional match chelate chromatography by using the Ni-NTA sorbent. The procedure to generate pure protein complexes with Gd^3+^ derived from gadolinium nitrate [Gd(NO_3_)_3_] was established taking into account the low solubility of Gd salts outside of pH 6.0–7.0.

The obtained complexes were tested for specific affinity to the U87MG human glioma and A375 human melanoma cell lines compared with normal human fibroblasts by using different methods including inductively coupled plasma atomic emission spectrometry (ICP-AES) for the Gd assay, fluorescent microscopy, and cell flow cytometry. Protein stability towards proteolysis was tested upon incubation with serum in vitro. Finally, in vivo experiments using C57BL/6 female mice bearing tumours caused by grafting a Ca755 murine adenocarcinoma were tested by using the E2-13W4, E1-W8, and E1-W12 proteins chemically labelled with cyanine 7 (Cy7). Taken together, our experimental data provide evidence about the robustness of the proposed technology to produce Gd^3+^-binding proteins; their specific tropism to human and murine tumour cells exemplified with U87MG human glioma, A375 human melanoma, and Ca755 murine breast adenocarcinoma; the high stability of the Gd–protein complexes in the bloodstream; and the low toxicity for cells and animals in the absence of exposure to radiation.

## 2. Results and Discussion

### 2.1. Protein Modelling

#### 2.1.1. Construction of the Basic Metal Binding Domain Containing the 4MBS-Domain

Gd^3+^ can tightly bind to calcium ion (Ca^2+^) binding sites of natural biomacromolecules because the radii of Gd^3+^ (0.99 Å) and Ca^2+^ (1.00 Å) are close [40,41,42,43], in addition to oxophilic ligand preferences [44]. Thereby, Gd^3+^ can interfere in intracellular calcium-dependent regulatory processes, an ability that is assumed to underlie the molecular mechanism of Gd toxicity [45,46]. Based on this, human calmodulin was used as a natural prototype of an artificial metal (i.e., Gd^3+^-binding protein domain). Directly using complete calmodulin in its natural form is impossible because this protein is involved extensively in cell-signalling regulatory networks. Therefore, the Ca^2+^-binding motif (D21-L33) was only used for de novo design of the Gd^3+^-binding domain. Previously [47], the Ca^2+^-binding loop from this amino acid sequence (D21–I28) was successfully used for complexation of Gd^3+^ within a short synthetic chimeric peptide. The metal binding domain requires the following: (1) an ordered structure, (2) an inert external surface of the native molecule compatible with the prokaryotic environment during protein synthesis and the eukaryotic environment when used for contrasting or therapy, and (3) rapid and efficient folding in *E. coli* upon the synthesis (cysteine residues should be avoided). With all these considerations in mind, ELP motifs were used as a domain-forming core. ELPs are a class of biopolymers that consist of repeated pentapeptide sequences Val-Pro-Gly-Xaa-Gly (VPGXG, where Xaa can be any common amino acid except proline), taken from human tropoelastin. This element has been used widely in synthetic biology to create biomaterials with various functions and properties [48]. Here, a metal-binding domain was constructed containing four identical metal-binding motifs from calmodulin alternating with ELP motifs of the composition 2 × (E(Y)-M-E-M)-E_end_, where E(Y) consists of two ELP repeats, one with Y (Tyr) and another with S (Ser) as Xaa; M is the metal binding site included in the D21–L33 portion of human calmodulin 1; E consists of four ELP repeats with S as Xaa; and Eend consists of two ELP repeats with S as Xaa. The serine residue bearing a polar neutral side chain was chosen as a guest Xaa in most ELP repeats to confer mild hydrophobic properties on one ELP unit (the grand average of hydropathicity [GRAVY] index is 0.2, computed by the EXPASy tool) to promote the formation of the hydrophobic core of the 4MBS-domain for overall hydrophilisation of the whole domain (the GRAVY index is −0.469, computed by the EXPASy tool) for solubility reasons.

Before designing an expression construct, the secondary structure of the whole 4MBS-domain was predicted by using trRosetta (Figure 1). The molecular modelling forecasted that the domain constructed by this method should have an ordered helix-coil conformation where most of the metal-binding site sequences are fixed in the coil conformation, which is beneficial for complexation with metal ions.

Next, the tertiary structure was modelled by trRosetta with restraints from both deep learning and homologous templates [49]. It predicted the possible spatial arrangement of the charged and hydrophobic regions within the 4MBS-domain. One of the predicted models is shown in Figure 2A. The domain has a globule-like shape harbouring a mild hydrophobic core surrounded by a hydrophilic envelope (the colour scheme is explained in the Figure 2 caption). The metal binding sites (red protein backbone) are located in the outer sphere. Similar results were obtained by trRosetta based on the de novo folding model, guided by deep learning restraints. Therefore, using ELP repeats provided the required globule-like structure of the designed MBS domain stabilised by hydrophobic interactions in the inner sphere.

#### 2.1.2. Construction of the RGD-Containing Region (3RGD-Domain)

A previous study provided examples of artificially created molecules containing several RGD motifs that are capable of simultaneously binding several integrin αvβ3 receptors, as confirmed by electron microscopy [50]. Closely spaced RGD-containing ligands immobilised on a substrate also cause integrin αvβ3 clustering [51,52,53]. A number of studies have also demonstrated that a monomeric RGD motif or a monoclonal antibody to integrin αvβ3 prevent clustering of these receptors [54]. Therefore, the RGD motif was repeated to promote clustering and internalisation of the designed protein into the target cells. Based on previous work [55], RGD was added in triplicate within the AVTGRGD motif separated from each other by a linker of 12 amino acids–5 × (SG)GS. The secondary structure of the whole 3RGD-domain predicted by trRosetta is shown in Figure 1.

#### 2.1.3. Linking the Designed Domains

F3 peptide has a large positive charge (pI 10.77) and does not have a stable conformation (the computed instability index is 90.34). The secondary structure predicted by trRosetta software includes mostly a coil conformation of the peptide bonds; an 8-amino-acid-long helix is the only regular element of F3 peptide (Figure 1).

Because the MBS and RGD domains were predicted to have a globular conformation, nucleolin-binding F3 peptide was used as a linker to connect two 4MBS-domains. Introducing F3 peptide to the designed protein of interest can putatively contribute to its ability to target tumours due to specific affinity to the surface nucleolin and simultaneously endows favourable physical and chemical properties such as hydrophilicity, stability, and neutral pI, among other features (Table 1). These parameters of the de novo designed protein become close to serum albumin that is likely beneficial for its longevity in the bloodstream. Moreover, F3 peptide is probably not involved in the formation of the MBS domain structure and accordingly the predicted model of whole proteins (Figure 2). Therefore, it remains entirely available for interaction with the cellular receptor.

With these considerations, F3 peptide and the 4MBS-domain were multiplied, with the latter repeated several times within the E1-W8 and E1-W12 constructs. This decision allowed endowing the constructs with favourable features like pI, stability, and GRAVY (Table 1), as was achieved in E2-13W4.

GFP was included in the MBS-protein molecule because its fluorescence allows a permanent visual tracing of its synthesis and distribution between the fractions along the overall experimental course at stages of the producer strain cultivation, protein purification, and uptake experiments. EmGFP was placed at the C-terminus of the fused protein to serve as a reporter of protein folding. A C-terminal location is most adequate for this role because acquiring the native conformation of the C-terminal portion of the protein depends on the formerly nascent N-terminal portion more than vice versa. Comparison of the forecasted 3D structures of the virtual MBS-proteins with and without EmGFP allowed hypothesising that the globule stability of GFP and the artificially designed parts of MBS-proteins are independent (data not shown). The physical and chemical properties of GFP are comparable to those of the artificially designed parts of MBS-proteins (Table 1).

Amino acid composition of the functional units of the designed proteins is summarised in Table 2.

### 2.2. Engineering DNA Constructs

The proposed modular composition of the Gd^3+^-binding protein platform requires development of a cloning technique allowing subsequent multiplication of short DNA repeats (MBS, RGD, ELP, and F3). Such DNA constructs often demonstrate instability in vivo and in vitro. Besides, the cloning scheme must render an optimal codon composition that would provide a high yield of the product when expressed in *E. coli*.

To design DNA oligonucleotides used for synthesis of the expression blocks, only the codons preferably used in *E. coli* were chosen (https://www.genscript.com/tools/codon-frequency-table, accessed on 12 February 2022). However, in view of the necessity to reduce the length of perfect direct repeats and therefore prevent in vivo instability of the constructs, different synonymic codons for Ser, Thr, and Leu were used. The length of the purchased oligonucleotides was limited to 63 nt to prevent possible single-nucleotide deletions that could appear due to incomplete condensation of phosphoimidates at each step. In addition, constructing tandem repeats with minor variations within the functionally uniform modules was required. Hence, the overall cloning scheme included stages of tandemisation of the repeats and the stages of joining of the established tandems.

The multiplexing principle used in this work is shown in Figure 3. This principle is based on restriction/ligation of an isocaudomers (i.e., pairs of restriction enzymes that have different recognition sequences but render compatible sticky ends upon cleavage) and uses the fact that upon homological ligation, the original site is retained, but upon heterological ligation, there are hybrid sites recognised by neither of the involved restriction enzyme. To realise this approach, a gene fragment to be multiplied must be flanked with different restriction sites of isocaudomers and the recipient plasmid must have a single site of isocaudomers. Simultaneous restriction and ligation in a single tube restores sites that appear due to ligation of the fragments in unwanted orientations. Therefore, these erroneous links are immediately broken, in contrast to the links between the fragments in the correct orientation. Hence, unlimited tandemisation of the repeats in a direct orientation can go on. The advantages of this technique are (1) rapid preliminary selection of co-directed insertion with single primer PCR (Figure 3, right) and (2) the possibility to use the enzymatic ligation assisted by nucleases ELAN principle for rapid tandem multimerisation of short repeats [56]. In some cases, depending on the sequence to be tandemised, seamless linking of protein motifs can be achieved; otherwise, a pair of amino acids encoded by a chimeric restriction site is added between the linked monomers. In this work, the BglII/BamHI pair was chosen, keeping in mind that the chimeric site encodes Gly-Ser. This amino acid motif is hydrophilic, does not impact the tertiary structure, and can serve as a part of the linkers between the protein domains.

The cloning scheme included the following main stages: (1) obtaining the MBS core sequence encoding the M motif (derived from human calmodulin) alternated with ELP repeats—the **B77end block**; (2) obtaining the nucleolin-specific F3-encoding unit with a peptide linker in the C-terminal position—**F3L block**; (3) combining the B77end block with the F3L block—the **W4 block**; (4) obtaining a unit encoding a triplicated RGD-derived motif from human fibronectin with a linker in the C-terminal position—the **RGD1(3) block**; (5) combining the RGD1(3) and W4 blocks within the pRSET-EmGFP expression vector—the **13W4 block;** (6) modifying a coding sequence just downstream of the translation initiation site of pRSET as described previously [57]—the **pE2-13W4 plasmid**; (7) engineering a plasmid with a duplicated W4 block—the **pE1**-**W8 plasmid**; and (8) engineering a plasmid with a triplicated W4 block—the **pE1-W12 plasmid**.

Thus, the plasmid construction scheme outlined in Figure 4 contains three stages suitable for successive tandem multiplying of the inserted fragment. Table 3 presents the multiplying potential of the initial plasmids.

All open reading frames (ORFs) were designed to exclude rare codons to ensure high recombinant protein yield. The GC content was also considered to facilitate engineering and protein yield. For practical reasons, however, it would be beneficial to further increase protein yield. Based on these considerations, the backbone of the p13W4 plasmid was modified slightly. It is known that varying the codon composition around the 5′-proximal region of messenger RNA (mRNA) could control the level of expression of the recombinant protein. Here, a predictive design method, UTR Designer [57], was used to determine the length and nucleotide composition of this region. In this way (online calculator), A and C nucleotides were preferable downstream from the ATG start codon of the 6His-tag in any composition with a total length of 15 nucleotides. Given the codon usage of *E. coli*, amino acids encoded by a combination of A and C, including an additional histidine adjacent directly to the 6His tag, were chosen. Codon frequencies were also considered to maximise protein production [58]. As a result, the site of the original pRSET plasmid (-cggggttct-) was replaced by the site (-accaccccacaacac-) that encodes Thr-Thr-Pro-Gln-His, which also led to replacement of the original 6His-tag to a 7His-tag. Two DNA variants of the insertion (E1 and E2) coding the same amino acid sequence were used. The above-mentioned procedure allowed increasing the desired protein yield by ~60% compared with the initial yield, as estimated by sodium dodecyl sulphate (SDS)-PAGE analysis of cell lysates, harbouring p13W4 (parental translation initiation site derived from pRSET), pE1-13W4 (E1 variant) or pE2-13W4 (E2 variant) (data not shown). Both variants provided a similar increase in protein yield, so they were not distinguished in the protein investigation experiments.

The motif composition of all three expression plasmids is summarised in Table 4.

### 2.3. Protein Purification

Evaluation of the expression conditions revealed that when cultivated under similar conditions, each protein yielded 40–50 mg per gram of biomass. Cultivation on nutrient-rich media (e.g., Terrific broth or 2 × YT) provided no or a low yield of the product. By contrast, slow growth in LB low-salt medium supplemented with glycerol as a carbon source provided the optimum protein yield in this study. Moreover, the oxygen supply was an important factor in both bacterial growth and recombinant protein synthesis and maturation. Therefore, maximum agitation was applied in the case of a small-scale cultivation (up to 50 mL) and forced artificial ventilation in the case of the large-scale cultivation (0.4–0.5 l). In addition, overnight gravity sedimentation at 4 °C was used to achieve maturation of the protein (including covalent automodification of GFP). Under these conditions, all three proteins were synthesised with a maximum yield in a soluble form (Figure 5).

The protein purification protocol includes convenient methods such as IEC and IMAC with additional emphasis on metal elimination. The purified proteins were identified by SDS-PAGE for protein size and purity (Figure 5B). The protein concentration was determined by the BCA protein assay and confirmed by densitometry with GelAnalyzer 19.1. The protein purity determined by densitometry varied in the range of 96–98%.

The molar extinction coefficient of all three proteins at 487 nm was similar; it ranged from 35,000 to 45,000 M^−1^ cm^−1^ depending on the cultivation condition and did not depend on the type of protein. The absorption spectrum of the E2-13W4 protein is shown in Figure 5C. The formerly reported molar extinction coefficient of the native Em-GFP at 487 nm was 57,500 (M^−1^ × cm^−1^) [59], which is just a little more than MBS-proteins.

### 2.4. Size Measurement

DLS measurements showed that as expected, the size of the putative monomeric form of MBS-proteins increased slightly alongside an increase in the molecular weight from E2-13W4 to E1-W8 to E1-W12 and reached approximately 100% by volume (Table 5 and Figure 6).

The particle size calculation used by DLS is based on Brownian motion of the particles. Despite the lower or almost equal molecular weight of E2-13W4, E1-W8, and BSA, the size of the latter is smaller (4.7 nm) than the former (9.2 and 10.7 nm, respectively). This fact probably reflects the disordered nature of MBS-proteins as suggested by the protein structure modelling. A disordered protein conformation renders a larger thermodynamic radius of the globule and therefore hinders its movement in solution. Experimental measurements indicated that MBS proteins are able to conserve the globule size corresponding to the presumed monomeric form under physiological conditions when incubated for 24 h at 37 °C in 0.15 M NaCl (pH 7.0). Complexation with Gd^3+^ did not induce statistically significant changes in the size of MBS proteins (data not shown).

### 2.5. Gd^3+^-Binding Affinity (mK_d_) and Binding Stoichiometry Determination

The equilibrium dialysis method [60,61,62] was used to determine the mK_d_ of the MBS–Gd^3+^ complex. This complex undergoes dissociation, see Equation (1), with a rate constant determined by Equation (2). The free Gd^3+^ concentration that penetrates the semipermeable membrane is the same in both chambers after equilibration and can be measured.
(1)$$MBS-Gd^{3+}↔MBS+Gd^{3+}$$
(2)$$K_{d}=\frac{\left[MBS_{free}\right]*\left[Gd_{free}^{3+}\right]}{\left[MBS-Gd^{3+}\right]}$$

The data obtained (Table 6) show that (1) the mean geometric dissociation constants of MBS–Gd^3+^ complexes in all three proteins are similar, so multimerisation does not affect the binding properties of the 4MBS domain; (2) the 4MBS domain provides a good Gd^3+^ binding capacity at physiological pH despite the presumed flexibility and disordered structure of the whole proteins. For comparison, the mKd of Ca^2+^ with calmodulin varies in the range of 3–20 µM [63].

For each protein, the MBS and Gd^3+^ binding stoichiometries were determined by the continuous variation method [64]. Briefly, a 50 μM solution of the Gd–XO complex in 20 mM HEPES (pH 7.0) was titrated with the corresponding protein in the same buffer. Displacement of Gd^3+^ from the Gd–XO complex was determined by a change in the absorbance at 570 nm [65]. Figure 7 shows the results of titration with the E2-13W4 protein. Similar results were obtained with the E1-W8 and E1-W12 proteins. For convenience of data processing, the optical density at 570 nm in Figure 7 is shown as the inverse. Thus, as expected, for all three proteins the number of bound Gd^3+^ ions corresponds to the number of binding sites in the protein molecule.

### 2.6. Relaxivity and MRI

The in vitro relaxivity values of the designed E2-13W4-Gd_4_, E1-W8-Gd_8_ and E1-W12-Gd_12_ complexes were measured at a field strength of 7 T to minimise the signal-to-noise ratio (SNR) at pH 7.0 with the following parameters: *TR* = 16,000; *TE* = 7.1; *TI* = 50, 100, 200, 400, 600, 800, 1000, 1500, 1980). Table 7 shows that all three protein-Gd^3+^ complexes exhibit similar r1 values, which are slightly higher than the r1 value of Magnevist^®^.

Figure 8 shows that, at 100 µM Gd^3+^, the designed protein complexes were able to introduce contrast enhancement in T1-weighted imaging at 1.5 T (a commonly used value in clinical practice). Indeed, the theoretically predicted protein structure exhibits low autonomous stability, which does not allow attaining high relaxivity of the MBS-Gd^3+^ complex. A higher relaxivity can be achieved by constructing a rigid frame around the metal binding site [16,19,66]. While there are approaches to achieve a rigid frame, they are not applicable in this work because non-natural amino acid blocks in the carrier proteins must be minimised to avoid an immune response upon intravenous administration to humans. The achieved relaxivity of the MBS-Gd^3+^ complexes is comparable to those of the linear chelator Magnevist^®^.

### 2.7. In Vitro Assessment of the Stability of MBS-Gd^3+^ Complexes in the Presence of Serum

The stability of a contrast agents and therapeutics in the bloodstream is one of the crucial factors for in vivo applications. In this study, the serum stability of the *de novo* synthesised Gd-binding proteins was determined by incubating them with 50% human serum at 37 °C for up to 72 h.

Surprisingly all three MBS-proteins in complexes with Gd^3+^ remained intact even after 72 h of incubation (Figure 9), indicating that the proteins are highly stable in blood.

### 2.8. Assay of MBS-Protein Uptake by Tumour Cells Using Fluorescence Microscopy

To confirm the ability of the obtained MBS-proteins to interact with (or enter into) tumour cells, in vitro experiments were carried out using the IN Cell Analyzer (fluorescence microscopy). The U87MG cell line was chosen as a candidate due to expression surface nucleolin [67,68], which can serve as a receptor for F3 peptide [69]. Moreover, U87MG expresses abundant integrin αvβ3 ([5.2 ± 1.4] × 10^4^ receptors/cell, or 18 ± 6 receptors/μm^2^) [70], which can bind the RGD peptide. The U87MG cell line allowed estimating the theoretical contribution of both RGD and F3 peptide elements to the overall specificity of the tumour markers when the affinity of E2-13W4, E1-W8 and E1-W12 were compared. All three proteins, alone or complexed with Gd^3+^, showed an excellent visual ability to enter U87MG cells. Figure 10 shows an example involving E2-13W4. The following observations were made. (1) There was no visual difference in the uptake level between the studied MBS-proteins upon incubation of the cells with the proteins at equal molar concentrations in the medium. In this regard, it is noteworthy that at an equal protein molar concentration, the average concentration of the RGD motif remains the same, but the F3 peptide concentration increases two and three times in the series E2-13W4 → E1-W8 → E1-W12 (Table 4). This fact may be explained by a high concentration of MBS-proteins in the medium, perhaps exceeding the level of saturation of the respective cell receptors, which eliminates the differences between them. (2) There was no difference in uptake upon complexation of proteins with Gd^3+^, a finding indicating that proteins designed in such a way can bind Gd^3+^ without affecting the targeting ability. (3) There was a slight increase in protein accumulation upon cultivation of the glioma cells in the presence of 2 µM Mn^2+^ (Figure 10, right), which supports the suggestion that integrin αvβ3 is involved in this process. Indeed, Mn^2+^ is a known powerful activator of the ligand-binding function of integrins [71,72]. Interestingly, 2 µM Mn^2+^ in the cell culture medium was sufficient to promote MBS-protein internalisation, while the previously used concentration of Mn^2+^ for the activation of integrins was significantly higher (0.5–2.0 mM).

### 2.9. Flow Cytometry

Flow cytometry was performed to assess the share of the cells interacting with the studied MBS-proteins. The U87MG human glioma cell line was again used because it showed good uptake of MBS-proteins in the previous experiment. The A375 cell line was also used because it has been demonstrated to express surface nucleolin (it selectively binds the AS1411 nucleolin-specific aptamer) [73,74]. Integrin αvβ3 [75] or integrins β1 and β3 [76] have also have been reported as the surface markers of A375 cells. Freshly prepared primary human PBMC were used as a normal blood cell reference. As shown in Figure 11: (1) both tumour cell lines exhibited a good ability to bind the studied MBS-proteins (experiments with the E1-W12 protein were carried out separately and revealed similar results); (2) most of the tumour cell population was involved in the interaction; (3) almost no PBMC bound MBS-proteins; and (4) the mean fluorescence intensity of A375 cells treated with MBS-proteins was somewhat higher than that of U87MG cells treated with MBS-proteins.

### 2.10. In Vitro Accumulation of Gd^3+^ Carried by the MBS-Protein in the Cells

Despite the encouraging results of experiments using the IN Cell Analyzer and flow cytometry, which showed a fast uptake of the studied MBS-proteins by tumour cells but not normal blood cells, direct data about ability of these proteins to bear Gd^3+^ were of interest. Therefore, comparative in vitro studies were performed to estimate the complexation of Gd^3+^ and MBS-proteins in U87MG cells, A375 cells and hTERT-immortalised human fibroblasts. ICP-AES was used for precise and sensitive measurement of Gd^3+^ after incineration. Commercial Magnevist^®^ contrast agent and Gd(NO_3_)_3_ salt solution were taken as references. All preparations were equilibrated by using the same Gd^3+^ concentration. A low Gd^3+^ concentration of 10 µM was chosen for in vitro binding experiments to adjust the concentrations of the contrast agent to those accessible for in vivo application, where using more carrier protein is not a cost-effective solution. The cells after incubation with the samples were harvested, quantified and incinerated, and the mineralised solution was analysed by ICP-AES to determine the Gd^3+^ concentration normalised per 10^6^ cells (Figure 12).

Statistical analysis of the data using two-way analysis of variance (ANOVA) led to the following conclusions. (1) The equilibrium Gd^3+^ concentration accumulated in each cell line after incubation with Gd(NO_3_)_3_ was ~6–12 higher than the concentration after incubation with Gd^3+^ organic complexes. However, the spontaneous uptake of Gd^3+^ does not exhibit specificity towards tumour cells. Complexation of Gd^3+^ with MBS-proteins reduces the penetration ability of Gd^3+^ but confers a tumour-specificity on this process. (2) Magnevist (a complex of Gd^3+^ with a linear ligand) exhibits very low or no specificity towards the tumour cell lines versus normal fibroblasts. (3) Compared with the normal fibroblast line, the U87MG and A375 cell lines exhibited significantly greater Gd^3+^ uptake for all three proteins but not for the references. The ratios of uptake by tumour and normal cells were 3.0–4.4 for U87MG cells and 2.3–2.6 for A375 cells (Table 8). (4) There was a small but statistically significant decrease in Gd^3+^ uptake by U87MG cells in the order E2-13W4 → E1-W8 → E1-W12. This pattern correlates with the decreasing RGD motif share in these proteins, while the F3 peptide share remains constant (Table 8). This observation suggests a predominant contribution of the RGD motif versus F3 peptide to deliver the MBS proteins to U87MG cells. (5) In A375 cells, however, there was no difference in Gd^3+^ uptake between the E2-13W4, E1-W8, and E1-W12 preparations. This observation suggests a predominant contribution of F3 peptide rather than the RGD motif to allow the protein vector to enter A375 cells.

Taken together, these observations show that all constructed MBS proteins tightly hold Gd^3+^ in the complex upon its interaction with cell receptors and internalisation. They efficiently prevent spontaneous uptake of free Gd^3+^ by the cells and thus confer a high tumour-directed specificity on its trafficking. Optimal doses and concentrations of the contrasting agents based on the constructed MBS proteins need to be elucidated by performing additional experiments.

Studies of the selective delivery of Gd to tumour cells have been and remain relevant to improve the technical development of neutron/photon capture therapy. Without the use of molecular delivery vehicles, the nonspecific uptake of Gd salt can reach very high values. Gd might be coordinated with ligands present in the incubation medium, for example, glucose [77]. To achieve specific delivery of Gd for neutron capture therapy or imaging applications, nanocomposites are used based on chitosan nanoparticles [78], liposomes [79], dendrimeric constructs [80], and single-walled carbon nanotubes [81], along with the study of new complexing agents [82].

### 2.11. Ex Vivo Imaging

To test the biocompatibility and tumour tropism of the developed MBS proteins, in vivo experiments were carried out. To this end, E1-W12 was chosen, containing the maximum number of vector modules per molecule among all obtained proteins. This protein was conjugated with a near-infrared (NIR) fluorophore, namely Cy7. This dye was chosen for its spectral properties, which provide the maximum penetration ability of the excitation light in the depth of the analysed tissue. The activated Cy7 dye reacts with amino groups of Lys residues, which are present both in EmGFP and F3 peptide. The latter reaction is undesirable, as it can hinder binding of F3 peptide to its cell receptor (nucleolin). To minimise the contribution of the unwanted Cy7 linkage, E1-W12 was chosen because it contains three copies of F3 peptides per molecule, not one or two as in E2-W4 and E1-W8, and the ratio of protein to be labelled to dye was set at 1:1 to 1:2.

C57BL/6 female mice were used in experiment as a syngeneic animal model for Ca755 murine adenocarcinoma [83,84]. We assumed that a rapid growth of Ca755 grafting induced tumours and, consequently, the formation of a tumour vascular network may favour binding of the analysed MBS protein with the tumour-specific receptor. Animals of each group were euthanised 1, 2, 4, or 24 h after E1-W12-Cy7 conjugate administration. A qualitative ex vivo evaluation of E1-W12 protein content in the liver, spleen, kidney, brain, blood, and induced tumour was performed with the IVIS^®^ Spectrum imaging system based on Cy7 fluorescence. Figure 13 and Figure 14 show the organs and the blood samples of the most representative animal from each time point. Figure 13 shows a set of organs taken from each individual animal and located in the same position (denoted with digits on the right), with blood samples in microtubes. Figure 14 shows the same organs and blood samples arranged in rows to assess the effect of accumulation over time.

The following observations were made from this experiment. (1) There was accumulation in the liver, kidney, and tumour. (2) It can be assumed that the circulation time in the blood is rather long, judging by the fluorescence of the blood samples, although it is difficult to estimate more accurately in this experiment. (3) None of the animals died and none exhibited any adverse reaction after administration of the E1-W12-Cy7 protein during the experiment. (4) After 24 h, there was a clear difference between the tumour itself and the surrounding tissue. When interpreting the results of an in vivo experiment testing the bio-distribution of complexes of the studied proteins with Gd, a relevant question is whether the distribution of Cy7 reflects the concentration and retention time of the protein in the tissue, because in the inner medium of the body the protein can degrade to a certain extent and release the dye, the pharmacokinetics of which will not coincide with the pharmacokinetics of the protein.

## 3. Materials and Methods

### 3.1. Materials

#### 3.1.1. Genetic Constructions

The CloneJET PCR Cloning Kit (Thermo Fisher Scientific, Waltham, MA, USA) was used to obtain intermediate constructs by direct PCR and restriction/ligation cloning of required genetic elements. The pRSET-EmGFP plasmid (Thermo Fisher Scientific) was used as an expression vector containing the GFP gene. High-fidelity restriction endonucleases BamHI-HF, BglII-HF, EcoRI-HF, NcoI-HF, and PstI-HF; T4-DNA-ligase; T4-polynucleotide-kinase; and shrimp alkaline phosphatase (rSAP) were purchased from New England Biolabs (Ipswich, MA, USA). The oligonucleotides were synthesised by the solid-phase method and purified by preparative polyacrylamide gel electrophoresis (PAGE) by Syntol LLC (Russia). Q5^®^ High-Fidelity DNA Polymerase (New England Biolabs) was used for all polymerase chain reaction (PCR). Ultrafree-DA Centrifugal Filter Units (Merck, Kenilworth, NJ, USA) were used for DNA extraction from agarose gel. The ZymoPURE™ Plasmid Miniprep Kit (Zymo Research, Irvine, CA, USA) was used for plasmid DNA purification. The authenticity of the plasmids was confirmed by Sanger sequencing performed by Eurogen CJSC (Russia). *E. coli* strain NiCo21(DE3) (#C2529H, New England Biolabs, Ipswich, MA, USA) was used for cloning and expression experiments according to the manufacturer’s protocol.

#### 3.1.2. Protein Purification

LB broth (Luria low salt, L3397-250G), lysozyme from chicken egg, AEBSF (A8456-25MG), DEAE–Sepharose^®^ CL-6B (CAS Number 57407-08-6), and other chemicals were obtained from Sigma-Aldrich (St. Louis, MO, USA). HisPur™ Ni-NTA Resin (88221), PD-10 gravity flow desalting column (G-25), and the Pierce™ BCA Protein Assay Kit (23227) were obtained from Thermo Fisher Scientific.

### 3.2. Software

SnapGene Viewer (GSL Biotech LLC, Chicago, IL, USA, available at snapgene.com) was used to design and manage plasmid maps. Oligo Analyzer 1.0.3, a free software program developed by T. Kuulasmaa, was used to design oligonucleotides. GelAnalyzer 19.1, a free software program developed by I. Lázár, was used for digital gel densitometry. Secondary protein structure prediction and three-dimensional (3D) modelling was carried out by using trRosetta [49]. Protein properties were predicted by using the EXPASy tool [85]. Cn3D4.3 software (available from https://www.ncbi.nlm.nih.gov/Structure/CN3D/cn3d.shtml, accessed on 12 February 2022) was used to design visual 3D models.

### 3.3. Cell Cultures

Human glioblastoma U87MG (ATCC^®^ HTB-14™), human malignant melanoma A375 (ATCC^®^ CRL-1619™), murine adenocarcinoma Ca755 (from the collection of NMRCO), and hTERT-immortalised BJ-5ta human fibroblasts (ATCC^®^ No. CRL-4001™) were kindly supplied by the cell collection of the Blokhin National Medical Research Center of Oncology of the Ministry of Health of the Russian Federation (Blokhin NMRCO). Human peripheral blood mononuclear cells (PBMC) were isolated from blood collected from a volunteer. Informed consent was obtained and approved at a meeting of the Local Ethics Committee of the VIGG (Protocol No. 2 dated 10 June 2020).

### 3.4. Methods

#### Genetic Engineering Manipulations

The scheme of plasmid assembly is shown in Figure 4. Four double-stranded DNA (dsDNA) blocks E(Y), M, E, and M* were obtained by pair annealing of synthetic oligonucleotides at 90 °C for 2 min followed by cooling to room temperature (RT) over 20 min. The oligonucleotide duplexes (0.1 pmol) were mixed and phosphorylated by T4 polynucleotide kinase, followed by ligation with T4 DNA ligase according to manufacturer’s protocol. The assembled product was used as a template for amplification with Bgl-E(Y) and Bam-M* primers containing BglII and BamHI flanking sites, respectively. The 190 base pair (bp) PCR product was used as an insertion DNA unit for cloning to pJET1.2blunt vector (step I, Figure 4) according to the user manual provided by the manufacturer. The resulting construct was denoted as pB7. The authenticity of the pB7 sequence was checked by Sanger sequencing.

To duplicate the E(Y)-M-E-M* block, the 190 bp PCR product from the previous step was digested with BglII and BamHI and ligated with the parental pB7 plasmid preliminarily digested with BamHI and dephosphorylated by rSAP (step II, Figure 4). A clone containing a duplicated E(Y)-M-E-M* insertion in a direct orientation was initially selected by negative selection by colony PCR with the single primer Bgl-E(Y), then by mapping with BamHI and PstI (a 716 bp fragment appeared). The resulting construct was denoted pB77. Its authenticity was checked by Sanger sequencing.

The Eend fragment encoding an additional ELP block was obtained by annealing synthetic oligonucleotides (End-for, End-rev). The oligonucleotides were mixed in the amount of 1 pmol each, annealed at 90 °C for 2 min and then cooled to RT over 20 min, resulting in a fragment with BamHI/BglII-compatible sticky ends. The fragment was ligated with the pB77 plasmid DNA preliminary digested with BamHI and dephosphorylated by rSAP (step III, Figure 4). The resulting construct was denoted pB77end. A clone containing an Eend insertion was selected by colony PCR with the pJET1.2 forward sequencing primer and End-rev (a 445 bp fragment appeared). Its authenticity was checked by Sanger sequencing.

A DNA unit including blocks F3 and L was obtained by annealing synthetic oligonucleotides F3-1-dir, F3-1-rev, FL-dir, and Bam-L-rev. Each oligonucleotide was phosphorylated separately by T4 kinase in the presence of 1 mM adenosine triphosphate (ATP). Oligonucleotides F3-1-dir, F3-1-rev, and FL-dir were mixed (0.1 pmol each); annealed at 90 °C for 2 min; and then slowly cooled to 58 °C with subsequent addition of 0.1 pmol of the oligonucleotide Bam-L-rev and then immediate cooling to 4 °C. The mixture was then supplemented with 0.5 mM dNTPs and 0.02 U/μL Q5-Polymerase and incubated for 2 min at 72 °C to allow single-stranded regions to be filled. The mixture was diluted, ligated by T4 ligase according to the manufacturer’s protocol, and used as insertion DNA for cloning into the pJET1.2blunt vector. The resulting plasmid was denoted pFL16 (step IV, Figure 4). Its authenticity was checked by Sanger sequencing.

pB77end was used as a template for PCR with the pJET1.2 forward sequencing primer and the pJET1.2 reverse sequencing primer. The 519 bp PCR product was re-precipitated to exchange buffer and digested with BglII and BamHI. The resulting 384 bp product was purified by extraction from a 1% agarose gel after electrophoresis and ligated with pFL16 preliminarily digested with BamHI and dephosphorylated by rSAP. The resulting plasmid was denoted pFLB77 (step V, Figure 4). The insertion block containing F3L-2(E(Y)-M-E-M*)-Eend elements was denoted W4.

The pW4 plasmid was also created by inserting the amplified W4 fragment of the pFLB77 plasmid into the pRSET-EmGFP vector at BglII/EcoRI sites in the same PCR/restriction/ligation reaction set as described below (step VIII, Figure 4) to verify protein product expression before further cloning.

The RGD-encoding DNA block was synthesised as a separate construct. First, two synthetic oligonucleotides (R1-for, R1-rev) were annealed by mixing 1 pmol of each at 90 °C for 2 min followed by cooling to RT for 20 min, subjected to one round PCR as described above and then used as a template in PCR with another pair of primers (Bgl-R1-for, Bam-R1-rev). The 73 bp PCR product flanked with BglII and BamHI sites was then digested with BglII and BamHI, purified by extraction from an agarose gel after electrophoresis and used as insertion DNA for cloning to an expression vector pRSET-EmGFP, preliminarily cut with BamHI and dephosphorylated by rSAP (step VI, Figure 4). Colony PCR with Bgl-R1-for and W4-seq-rev primers were used to determine both the presence and the direct orientation of the insert. The resulting positive plasmid denoted pRGD1(1) was used for subsequent insertion of the same 73 bp PCR product digested with BglII and BamHI at the BamHI site. The abovementioned procedure was repeated two more times to obtain the pRGD1(3) plasmid containing three tandem RGD motifs located upstream to EmGFP (step VII, Figure 4). Colony PCR with one Bgl-R1-for primer was used for rapid pre-selection of colonies with co-directed insertions. The authenticity of pRGD1(3) was verified by Sanger sequencing.

To combine the RGD-encoding block with the W4 block (F3L-2(E(Y)-M-E-M*)-Eend), the W4 gene fragment was amplified by PCR with primers (Bgl-F-for, Eco-W4-rev) and pFLB77 as a template, rendering a 536 bp product. This product was purified by preparative agarose gel electrophoresis, digested with BglII and EcoRI and ligated with the pRGD1(3) plasmid cut with BamHI and EcoRI (step VIII, Figure 4). The resulting construct was denoted p13W4. Its authenticity was checked by Sanger sequencing carried out by Eurogen LLC (Russia).

To produce a plasmid providing elevated protein yield, p13W4 was used as a template for a whole-plasmid PCR with preliminary phosphorylated primers Exp2-for and Exp-rev (or Exp1-for and Exp-rev). The 4290 bp PCR product was ligated and used for transformation of *E. coli*. The resulting plasmid was denoted pE2-13W4 (or pE1-13W4) (step IX, Figure 4). Pre-selection was performed by colony PCR with primers Exp2-seq-for and W4-seq-rev (or Exp1-seq-for and W4-seq-rev), resulting in a 901 bp fragment. The authenticity of the pE2-13W4 and pE1-13W4 plasmids was checked by Sanger sequencing.

To multiply the W4 unit, the p13W4 plasmid was used as a template for PCR with primers Eco-F-for and Eco-W4-rev. A 536 bp product flanked with EcoRI restriction sites was obtained. It was purified by extraction from an agarose gel and cut with EcoRI. The purified fragment was ligated with the pE1-13W4 plasmid preliminarily digested with EcoRI and dephosphorylated by rSAP (step X, Figure 4). The resulting construct was denoted pE1-W8. Its authenticity was checked by Sanger sequencing.

The 540 bp long product obtained by PCR of the p13W4 plasmid with primers Nco-F-for and Nco-E-rev cut with NcoI was ligated with the pE1-W8 plasmid preliminary digested with NcoI and dephosphorylated by rSAP (step XI, Figure 4). The resulting construct was denoted pE1-W12. Its authenticity was checked by Sanger sequencing.

Plasmid maps and oligonucleotide sequences are listed in Appendix A.

### 3.5. Protein Purification

#### 3.5.1. Culture Growth

A single colony of *E. coli* NiCo21(DE3) freshly transformed with the pE2-13W4, pE1-W8, or pE1-W12 plasmid was inoculated into 500 mL of the cultivation medium (Luria low-salt broth supplemented with 0.2% glycerol and 100 μg/mL ampicillin). The cultivation was carried out in a hermetically closed 2.5 L plastic cylinder (20 cm in diameter) with forced air ventilation through inlet and outlet tubing at 442 L/min and agitation at 100 rpm for 19–22 h at 37 °C. The medium volume reduced twofold under these conditions.

The culture was then incubated overnight in a glass cylinder at 4 °C for spontaneous sedimentation of the cells. The concentrated cell suspension was then eventually precipitated by centrifugation at 10,000× *g* for 1 min. The yield of the wet cell biomass was 8.4 ± 0.8 g/L relative to the initial medium volume.

#### 3.5.2. Cell Lysis and Clearing the Lysate

The cells were suspended in 10 mL of 2 mM EDTA. Sodium hydroxide was added to the suspension at a final concentration of 10 mM, and then the suspension was agitated for 2 min and neutralised by adding concentrated Tris pH 8.0 to a final concentration of 100 mM. Then, AEBSF (up to 2 mM) and lysozyme (0.02 mg/mL) were added, and the suspension was incubated for 20–30 min at 37 °C. The resulting suspension of protoplasts was subjected to homogenisation by sonication for 5–7 rounds under the following alternating regimes: 20 s treatment at 24–25% of the maximum followed by 25 s chilling on ice. The lysate was subjected one to two times to freezing/thawing and clarified by centrifugation at 20,000× *g* for 15 min at 4 °C. The pellet was discarded; polyethylenimine was added to the supernatant to a final concentration 0.2%, and the solution was incubated for 10 min at RT and centrifuged at 20,000× *g* for 15 min at 4 °C. The supernatant was collected, filtered through a 0.22 µm syringe cartridge, and used for further purification.

#### 3.5.3. Ion-Exchange Chromatography

Three millilitres of DEAE–Sepharose^®^ CL-6B were used to process a soluble protein fraction obtained from 8–10 g of crude biomass. The resin was equilibrated with IEC loading buffer (50 mM Tris [pH 8.0], 0.05 M NaCl). The cleared lysate from the previous step was applied on the column and washed with 30 mL of IEC loading buffer. Desorption of the protein of interest was carried out in IEC elution buffer (50 mM Tris [pH 8.0], 0.5 M NaCl). The collected eluate was used directly for the subsequent purification step.

#### 3.5.4. Metal-Chelate Chromatography

Five millilitres of the HisPur™ Ni-NTA Resin were used. The resin was equilibrated with immobilised metal affinity chromatography (IMAC) loading buffer (50 mM Tris [pH 8.0], 0.3 M NaCl). The eluate from 3 mL of DEAE–Sepharose^®^ CL-6B was applied on the column, washed with 30 mL of the same buffer, and eluted with IMAC elution buffer (50 mM Tris [pH 8.0], 0.5 M imidazole).

#### 3.5.5. Metal Ion Elimination

The eluate from IMAC (total volume 10 mL) was supplemented with EDTA to a final concentration of 10 mM and dialysed several times against 1 L aliquots of the following series of dialysis buffers: (1) 50 mM Tris (pH 7.5), 10 mM EDTA, 0.02% sodium azide (NaN_3_)–once; (2) 50 mM Tris (pH 7.5), 0.02% NaN_3_–twice; and (3) 50 mM Tris (pH 7.5)–once. Each dialysis lasted for 24 h at 4 °C.

#### 3.5.6. Desalting the Protein

Gel filtration on PD-10 desalting columns packed with Sephadex G-25 resin was used to transfer the pure proteins of interest to 10 mM HEPES (pH 7.0). The pure proteins were aliquoted and stored at −20 °C until needed.

### 3.6. Size Measurement

Dynamic light scattering (DLS) measurements were performed with the Zetasizer Nano ZS (Malvern Instruments, UK) with laser 633 nm and θ = 173° at 25 and 37 °C in 1 cm quartz cuvettes (Z276669-1EA, Sigma-Aldrich). Samples were taken in buffer (10 mM HEPES [pH 7.0], 0.15 M NaCl) and analysed with standard settings. The data were processed via the ‘high resolution analysis’ function of the Malvern software. All measurements were performed independently three times, and the data were averaged.

### 3.7. Gd^3+^-Binding Affinity Determination

The equilibrium dialysis method was used to determine the geometric mean of the dissociation constant (mK_d_) of the MBS–Gd complex [60]. Protein–Gd complexes were prepared by mixing 2 mL of each protein in 10 mM HEPES (pH 7.0) with a stock aqueous solution of Gd(NO_3_)_3_ at RT to obtain an equimolar ratio of MBS and Gd (492.6 µM). Solutions were then put in dialysis devices (Slide-A-Lyzer™ MINI Dialysis Device, 10K MWCO, 2 mL, 88,404, Thermo Fisher Scientific) and dialysed against 2 mL of 10 mM HEPES (pH 7.0) (denoted ‘out’) for 24 h at 25 °C and 50 rpm to reach full equilibrium. The absence of nonspecific losses of Gd during the experiment was confirmed by preliminary dialysis with 2 mL of Gd(NO_3_)_3_ aqueous solution at the same conditions. The concentration of free Gd^3+^ in the dialysis buffer was determined spectrophotometrically [86]. Briefly, 25 µM xylenol orange (XO) indicator was added to 2 mL of the ‘out’ dialysis buffer and the Gd^3+^ concentration was determined by using a calibration curve of the standard Gd-XO solutions at the same conditions. The experiment was repeated three times. The mK_d_ of the MBS–Gd complex of each protein was calculated by using Equation (3).
(3)$$mK_{d}=\frac{\left[Gd^{3+}\left(free\right)\right]\left[MBS\left(free\right)\right]}{\left[MBS:Gd^{3+}\right]}$$

### 3.8. Binding Stoichiometry Determination

The binding stoichiometry of MBS and Gd^3+^ were determined by the continuous variation method [64] for all three proteins. Briefly, a solution of the 50 μM Gd–XO complex in 20 mM HEPES (pH 7.0) was titrated with one of the proteins in the same buffer. Displacement of Gd^3+^ from the Gd–XO complex was determined by change in absorbance at 570 nm [65]. Titration was carried out at RT in a quartz cuvette with an optical path length of 10 mm using a Varian Cary 50 UV-Vis spectrophotometer (Agilent, Santa Clara, CA, USA).

### 3.9. Relaxivity Measurement

*T*_1_ relaxivity of the obtained complexes and Magnevist^®^ were calculated per [Gd^3+^] as described by Tamura et al. [87] A set of images of each sample (concentration 10–100 µM Gd^3+^ in 0.3 mL of 10 mM HEPES [pH 7.0]) was obtained by using an inversion-recovery sequence with the following parameters: *TR* = 16,000; *TE* = 7.1; *TI* = 50, 100, 200, 400, 600, 800, 1000, 1500, and 1980 (17 °C, 7T, Bruker BioSpin, ClinScan, Ettlingen, Germany). The signal intensity (*SI*) of each sample at different *TI* values was measured using ImageJ Software (National Institutes of Health, Bethesda, MD, USA). Curves of *SI* dependence on *TI* for each concentration were constructed, and the *T*_1_ relaxation time was determined by using Mathcad approximation (Needham, MA, USA). The dependence of reverse *T*_1_ relaxation time on the Gd^3+^ concentration was plotted, and *T*_1_ relaxivity was calculated as the tangent of the inclination angle using Equation (4).
(4)$$SI=k*[1-\left(1-\cos\left(f\right))-e^{-\frac{TI}{T1}}-2\cos\left(f\right)*e^{-\frac{TR-\frac{TE}{2}}{T1}}+\cos\left(f\right)*e^{-\frac{TR}{T1}}\right]$$

### 3.10. MRI of Phantoms

T1-weighted MR images of phantoms were acquired at 1.5 T and ambient temperature in the appropriate buffer. The procedure used 3D spoiled gradient-recalled echo sequence (SPGR) with *TR* = 500; *TE* = 10; and a = 40°.

### 3.11. Assessment of Stability of MBS-Proteins in Complexes with Gd^3+^ in the Presence of Serum In Vitro

The Gd^3+^ complex with MBS proteins encoded by the pE2-13W4, pE1-W8, or pE1-W12 plasmid was freshly prepared in 10 mM HEPES (pH 7.0) (MBS:Gd^3+^ molar ratio was 1:1); incubated with 50% human serum over 0, 24, or 72 h at 37 °C; and subjected to native PAGE. Gel images were acquired and analysed by laser scanning the gels with the Typhoon FLA 7000 instrument (GE Healthcare, Chicago, IL, USA); GFP fluorescence intensity of the band corresponding to the untreated proteins in all samples was measured to estimate the protein degradation.

### 3.12. Assay of MBS-Proteins Uptake by Tumour Cells by Fluorescence Microscopy

The U87MG glioblastoma cell line was cultured in 6-well cell culture plates in a 5% CO_2_ atmosphere at 37 °C. Upon reaching 80% confluence, the proteins were added to the medium at a final concentration of 5 µM (1/20 of the final volume) and incubated for 24 h in a 5% CO_2_ atmosphere at 37 °C. One well was incubated with the proteins alone and another well was supplemented with manganese (II) chloride (MnCl_2_) at a concentration of 2 µM. The cells were then stained with 4′,6-diamidino-2-phenylindole (DAPI) (D1306, Invitrogen, Waltham, MA, USA) for 5 min, washed twice with cold (4 °C) phosphate-buffered saline (PBS), and analysed by using the IN Cell Analyzer 6000 instrument (Cytiva, Marlborough, MA, USA).

### 3.13. Flow Cytometry

The U87MG human glioma and A375 human malignant melanoma cell lines and freshly prepared human PBMC were used. The U87MG and A375 cells were plated at a density of 5 × 10^5^ cells per well in a 6-well cell culture plate and incubated in 10% RPMI for 24 h in a 5% CO_2_ atmosphere at 37 °C. Human PBMC from a healthy volunteer were isolated with a BD Vacutainer^®^ CPT™ Mononuclear Cell Preparation Tube (#362753) according to the manufacturer’s protocol and cultured in the same conditions. Proteins were added to the tumour cells and lymphocytes at a concentration of 5 µM and incubated for 24 h in a 5% CO_2_ atmosphere at 37 °C. The cells were then washed twice with PBS, detached with Versene Solution (#15040066, Thermo Fisher Scientific), and analysed by flow cytometry (NovoCyte, ACEA Bioscience, San Diego, CA, USA).

### 3.14. In Vitro Accumulation of Gd^3+^ in Cultured Cells

The U87MG human glioma and A375 human malignant melanoma cell lines were used alongside human fibroblasts (as a reference). E2-13W4, E1-W8, and E1-W12 complexes with Gd^3+^ were prepared by mixing the corresponding protein with the stock solution of Gd(NO_3_)_3_ in 10 mM HEPES (pH 7.0) with a MBS:Gd^3+^ molar ratio of 1:1. An aqueous solution of Gd(NO_3_)_3_ and commercial Magnevist (Bayer Pharmaceuticals AG, Germany) MRI contrast agent were used as reference samples. The cells were plated at a density of 5 × 10^5^ cells per well in a 6-well cell culture plate and incubated in 10% RPMI for 24 h in a 5% CO_2_ atmosphere at 37 °C. The culture medium was then replaced with fresh medium, and the Gd^3+^ concentration was adjusted to 10 µM in each well by using the prepared complexes alongside the reference samples; the cells were then incubated for another 24 h. The cells were then washed twice with 2 mL of Dulbecco’s PBS (NaCl 137.0 mM, KCl 2.7 mM, Na_2_HPO_4_ 8.1 mM, KH_2_PO_4_ 1.5 mM, pH 7.4). After counting the number of the cells, they were harvested and used for Gd^3+^ analysis. Briefly, cell pellets containing a known number of the cells were suspended in 50 μL of concentrated nitric acid and incinerated by heating at 60 °C for 2 h; the solutions were diluted 100 times or more with deionised water and filtered through a 0.45 µm Millex filter unit (Merck). Determination of the average weight concentration of Gd^3+^ was carried out with a ICP-AES (4200 MP-AES, Agilent, Santa Clara, CA, USA) using a calibration graph. For this, standard Gd^3+^ solutions (500, 1000, 1500, and 2000 ppb) were prepared by diluting the Gd standard solution (1 mg/mL) in 2% (w/w) nitric acid (HNO_3_) for ICP (Merck). A 2% aqueous solution of HNO_3_ was used as a blank. All experiments were carried out in triplicate. The Gd^3+^ concentration was recalculated per 10^6^ cells; a standard error was calculated as the variation between three parallel tests within the replicate.

### 3.15. Ex Vivo Imaging

#### 3.15.1. Labelling MBS-Proteins with Cy7 Fluorescent Dye

MBS-proteins were labelled with Cy7 fluorescent dye according to a standard protocol [88]. Briefly, 5 mL of E1-W12 (100 μM in 10 mM Tris buffer, pH 8.0) was mixed with 250 μL of cyanine 7-*N*-hydroxysuccinimide ester (4 mM in dimethyl sulphoxide [DMSO]) and incubated for 4 h at RT in the darkness. The labelled protein was purified from low-molecular-mass impurities by size-exclusion chromatography on prepacked PD-10 (Sephadex G-25) column equilibrated with PBS (pH 7.4) according to the manufacturer’s instructions. The protein conjugate was filtered through a 0.22 μm syringe filter for sterilisation and elimination of aggregates. The protein and Cy7 concentrations in the final solution were determined by spectrophotometry and confirmed by denaturing PAGE.

#### 3.15.2. Animal Experiment

Female C57BL/6 mice (20–22 g, 8–9 weeks old) were purchased from Pushchino Nursery (Russia). The mice were kept in standard cages at 21 °C. The photoperiod was 12 h of light and 12 h of dark per day. The animals had access to standard laboratory feed and water ad libitum. Before the experimental procedures, the mice were anaesthetised by intraperitoneal injection with fentanyl (0.05 mg/kg body weight), midazolam (5 mg/kg body weight), and medetomidine (0.5 mg/kg body weight) in 0.9% sodium chloride solution as described previously [89].

Tumour grafting was performed by subcutaneous inoculation of mice with a culture of the Ca755 breast adenocarcinoma permanent cell line on the right hind paw (50 μL of a 14% tumour cell suspension in Hanks’ Balanced Salt Solution [HBSS]). Four experimental groups of six animals each were formed once the tumours attained an average volume of 100–150 mm^3^. Two hundred microlitres of E1-W12-Cy7 conjugate (corresponding to 1.27 mg [15 nmol] of E1-W12 and 5 µg [10 nmol] of Cy7) in PBS was administered intravenously. The animals were euthanised 1, 2, 4, or 24 h after E1-W12-Cy7 conjugate administration. Subsequently, qualitative ex vivo evaluation of E1-W12 protein content in the tumour, liver, spleen, kidney, brain, and blood was performed with the IVIS^®^ Spectrum imaging system (Perkin Elmer, Waltham, MA, USA) based on the Cy7 fluorescence.

In vivo experiments including intravenous administration of the tested MBS protein samples labelled with fluorescent dye and euthanasia were carried out in accordance with the ethical requirements of the European Convention for the Protection of Vertebrates used for Experimental and other Scientific Purposes (ETS N123; adopted in Strasbourg 18 March 1986). The experimental protocol was approved at a meeting of the Local Ethics Committee of the VIGG (Protocol No. 2 dated 10 June 2020).

## 4. Conclusions

Recombinant proteins have traditionally been perceived as a product with a high manufacturing cost in comparison to products of chemical synthesis. Moreover, their application for in vivo theranostics for BRT/MRI requires a thorough purification of the substance to obtain endotoxin-free products. The recombinant proteins often demonstrate instability in vivo, and products of their degradation are accumulated in the liver and kidney. In the case of the Gd^3+^-binding preparation, this may cause intoxication of the liver and kidney tissue with the heavy metal ion. By contrast, synthetic cyclic ligands, for example, DOTA, traditionally used for manufacturing the currently used BRT/MRI agents firmly retain Gd^3+^ until its excretion in the urine or bile. Recombinant proteins have a high molecular mass compared with the popular cyclic chelators; therefore, the mass load of the recombinant protein for delivery of a molar units of Gd^3+^ is usually high, a factor that increases the already high manufacturing cost of the contrast agent. Taken together, these considerations have generated scepticism towards the idea of producing BRT/MRI contrast agents based on recombinant proteins. Our work provides evidence of highly efficient protein carriers to solve this task.

First, a challenging task of establishing a genetically stable construct made of multiple tandem repeats was solved. This was achieved by randomly tandemising functionally uniform but structurally divergent blocks containing synonymous codons or interchangeable amino acid residues in calmodulin modules and ELP repeats. Tandemisation of the initial blocks was carried out by a restriction/ligation procedure of the ‘semi-compatible’ sticky ends formed by BglII and BamHI restriction enzymes with subsequent PCR of the polymerised repeats and in vivo selection of the cloned sequences in *E. coli*. As a result, the W4 block with a high Gd^3+^ binding capacity due to multiple binding sites was established. It had optimal chemical and physical properties denoted by the optimal ratio between calmodulin modules with an acidic pI and cationic F3 peptide. The resulting pI somewhat below 7.0 and moderate hydrophobicity conferred by the ELP repeats provided efficient folding and a stable globular structure of the artificial domain, with a negligible or none share of α-helices and β-strands in its secondary structure. Optimising codons near the translation initiation site of the pRSET-GFP vector provided a high yield of soluble protein in *E. coli*. The optimal pI reduced losses at the stage of the protein complex formation with Gd(NO_3_)_3_, which is only soluble in a narrow pH range.

The use of multiple tumour-specific ligands (three RGD copies per molecule alongside one to three copies of F3 peptide) provided a high selectivity towards the tested tumour cells (U87MG human glioma and A375 human melanoma) compared with the normal human fibroblasts that were independently proved by three methods (fluorescent microscopy, fluorescent flow cytometry, and metal assay in cells with ICP-AES). The proteins demonstrated high proteolytic stability towards the human blood plasma proteases, an important factor for long-term maintenance of the contrast agent in the bloodstream.

Finally, in vivo trials demonstrated an optimal biological distribution of the protein in mice. A protein conjugated with Cy7 accumulated in the liver, kidney, and tumour, but not in other healthy tissues. This observation suggests optimal recognition of even small tumour foci in most tissues including breast, lung, and ovary cancer; gliomas and neuroblastomas; metastases of melanomas; carcinomas of the stomach, intestine and colon; sarcomas; and lymphomas. Taken together, our study proves that a complex approach to designing Gd-binding proteins, including optimising the number of the metal binding sites and tumour ligand peptides and using an efficient biosynthesis and purification procedure, allows overcoming major limitations of the recombinant technology for producing BRT/MRI contrast agents. Moreover, this approach enhances the advantages of these agents, including long-term stability in the bloodstream and a high specificity towards small tumour foci, factors that are pivotal for early diagnosis and efficient elimination of curable tumours at early stages of their development.

## Figures and Tables

**Figure 1 ijms-23-03297-f001:**
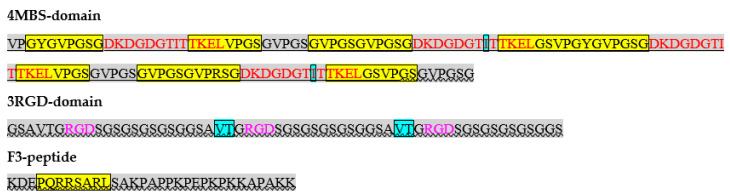
Putative secondary structure of the 4MBS-domain, 3RGD-domain, and F3-peptide predicted by trRosetta. MBSs are shown in the red font, α-helices are shown in boxes with yellow highlighting, β-strands appear in boxes with blue highlighting, and the residues in the coil conformation are highlighted with grey. The residues underlined with a straight line are putatively fixed in an ordered conformation, while the residues underlined with a wavy line have a disordered conformation.

**Figure 2 ijms-23-03297-f002:**
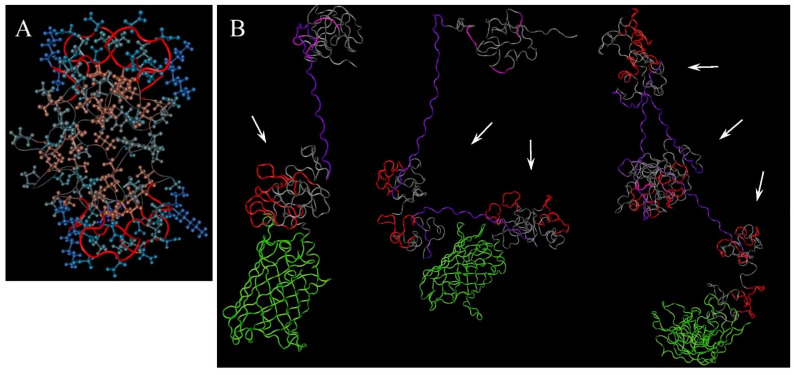
(**A**). A graphical presentation of the 4MBS-domain generated by Cn3D4.3. The protein sidechains and backbones are coloured based on the hydrophobicity scheme* of Cn3D4.3 with the exception of MBSs, the backbones of which are shown in red. (**B**). The putative tertiary structure of the designed proteins E2-13W4, E1-W8, and E1-W12 (from left to right) predicted by trRosetta with restraints from both deep learning and homologous templates. The graphical presentation was generated by Cn3D4.3. MBSs are shown in red, F3 peptide is shown in purple, RGD is shown in magenta, and EmGFP is shown in green. The arrows indicate the 4MBS-domains. * *Protein backbones and sidechains are coloured according to their free energy of immersion in water. Blue is hydrophilic and red is hydrophobic*.

**Figure 3 ijms-23-03297-f003:**
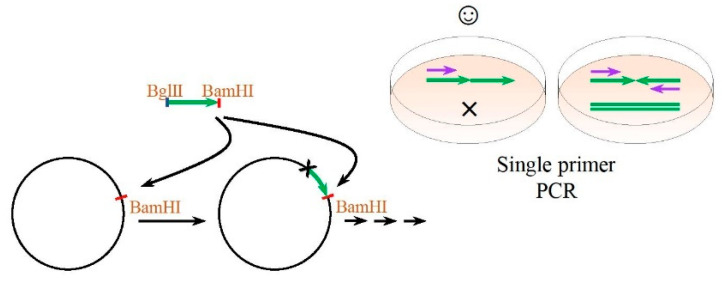
Principle scheme of cloning tandem repeats based on restriction/ligation of isocaudomers used in this work.

**Figure 4 ijms-23-03297-f004:**
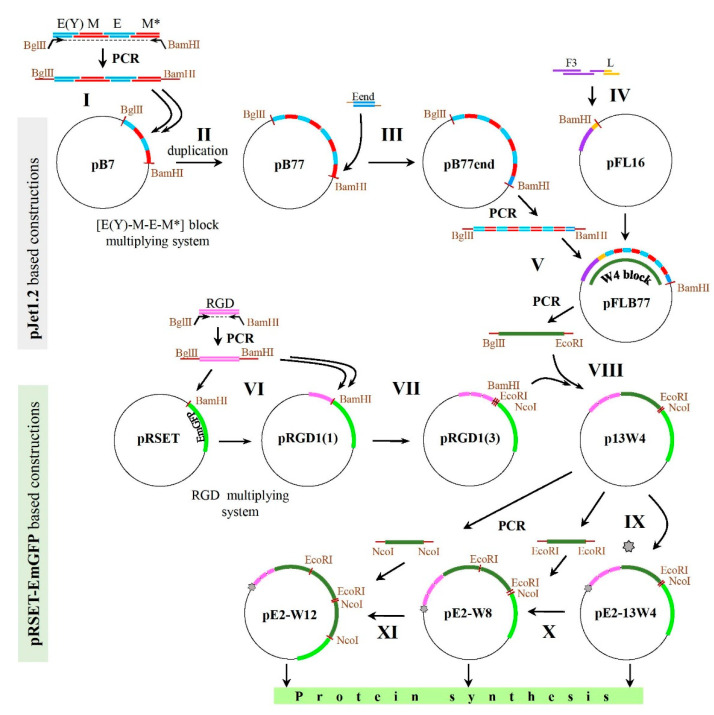
Overall scheme for constructing the plasmids used in the work. All constructs are based on the pJET1.2 vector backbone (at the top) or the pRSET-EmGFP backbone (at the bottom). The colour codes are used to delineate the following amino acid sequences: E(Y)-[VPGYG-VPGSG], E-[(VPGSG)_4_], M and M*-[DKDGDGTITTKEL], Eend-[(VPGSG)_2_], F3-KDEPQRRSARLSAKPAPPKPEPKPKKAPAKK, L-[(G_4_S)_2_(G_3_S)_1_], and RGD-AVTGRGD(SG)_5_GS. The enhanced expression peptide tag is marked with a grey star.

**Figure 5 ijms-23-03297-f005:**
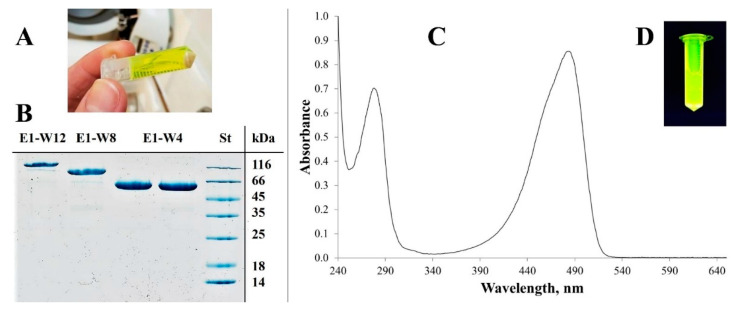
(**A**) E2-13W4 within a crude *Escherichia coli* lysate after sonication and subsequent centrifugation. (**B**) Analysis of the homogeneity of the purified MBS-proteins with 15% SDS-PAGE with subsequent staining of the gel with Coomassie brilliant blue R-250. (**C**) Absorption spectrum of the purified E2-13W4 protein (1.2 mg/mL) in 10 mM HEPES (pH 7.0). (**D**) The same solution of the E2-13W4 protein exposed to 314 nm ultraviolet light.

**Figure 6 ijms-23-03297-f006:**
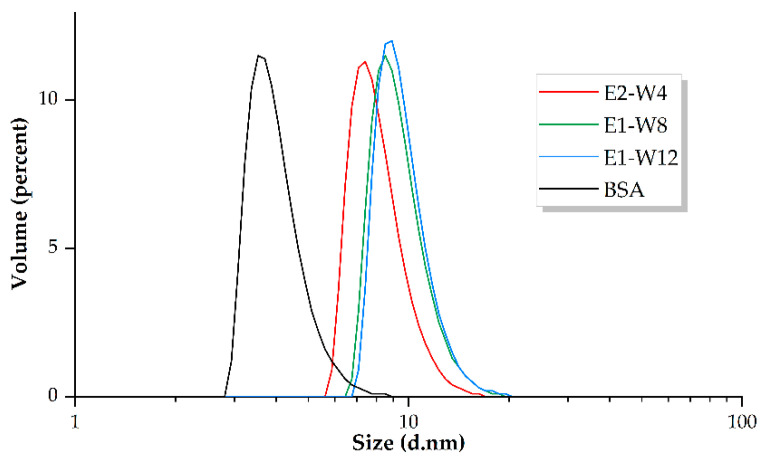
Size distribution of MBS proteins by volume measured by DLS. Red—E2-13W4; green—E1-W8; blue—E1-W12; black—BSA.

**Figure 7 ijms-23-03297-f007:**
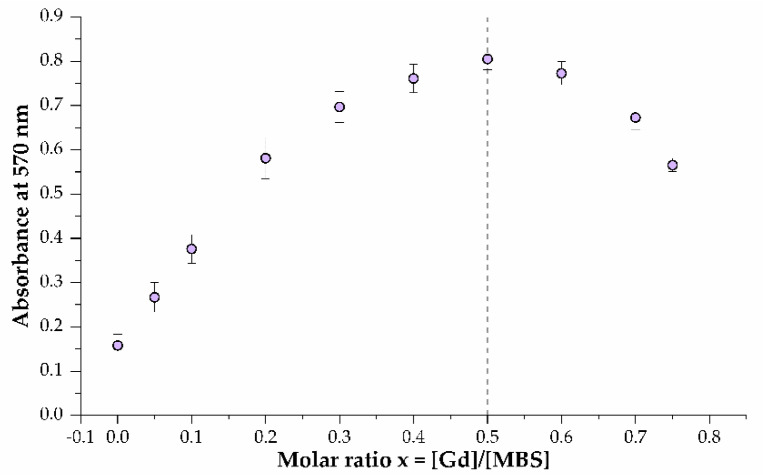
Job graph for Gd^3+^ binding to the E2-13W4 protein. The observed maximum at x = 0.5 corresponds to a stoichiometric ratio of 1:1. The results are represented as arithmetic means, and the error bars represent standard deviations (*n* = 3).

**Figure 8 ijms-23-03297-f008:**
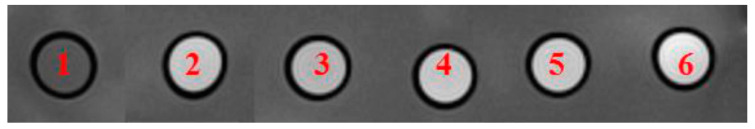
T1-weighted MRI of phantoms were made at 1.5 T, room temperature in correspondent buffer (10 mM HEPES, pH 7.0) and 100 µM Gd^3+^ concentration. 3D SPGR TR = 500, TE = 10, a = 40°. (1)–H_2_O; (2)–Gd(NO_3_)_3_; (3)–Magnevist^®^; (4)-E2-13W4-Gd_4_ complex; (5)–E1-W8-Gd_8_ complex; (6)-E1-W12-Gd_12_ complex.

**Figure 9 ijms-23-03297-f009:**
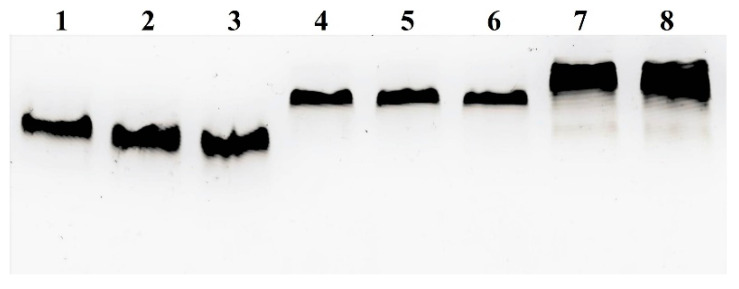
Native 12% PAGE of protein-Gd^3+^ complexes. Lanes: 1–3, E2-13W4; 4–6, E1-W8; 7 and 8, E1-W12. Lanes 1, 4 and 7 are samples before incubation; lanes 2 and 5 are 24-h incubation with serum; and lanes 3, 6 and 8 are 72-h incubation with serum.

**Figure 10 ijms-23-03297-f010:**
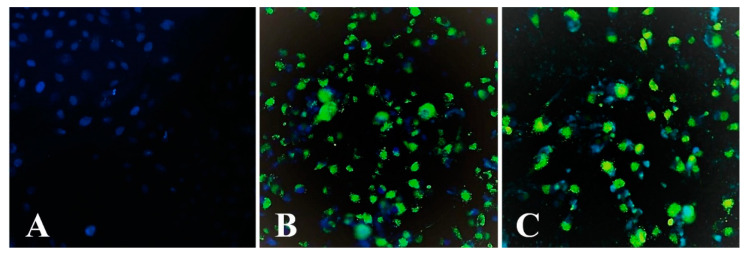
Uptake of the Em-GFP (**A**), E2-13W4 protein (**B**,**C**) by U87MG cells during 24-h incubation without (**B**) and with (**C**) 2 µM Mn^2+^. The green colour indicates protein and the blue colour indicates cell nuclei stained with DAPI.

**Figure 11 ijms-23-03297-f011:**
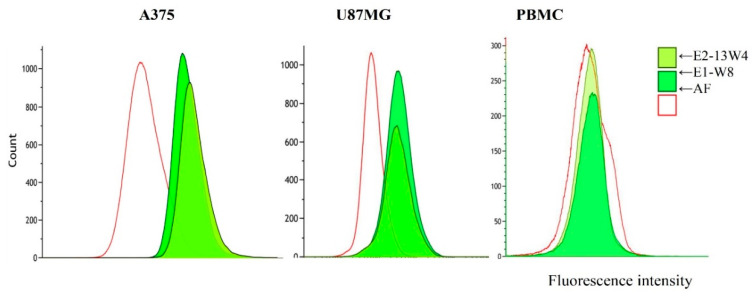
Histograms showing fluorescence intensity at 530 nm of A375 cells, U87MG cells and human PBMC before (autofluorescence [AF], red line) and after 24-h incubation with the E2-13W4 or E1-W8 protein (green filled curves).

**Figure 12 ijms-23-03297-f012:**
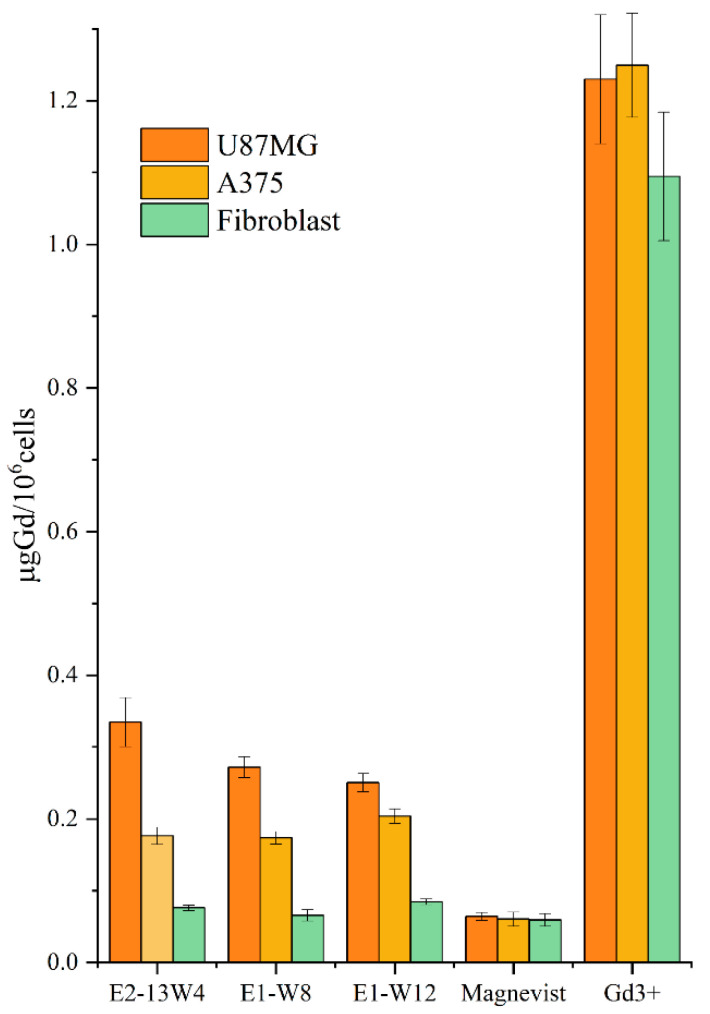
Gd^3+^ weight content per 10^6^ cells after 24 h incubation with the studied MBS proteins and control samples determined by ICP-AES. The results are presented as arithmetic means, and the error bars represent standard deviations (*n* = 3).

**Figure 13 ijms-23-03297-f013:**
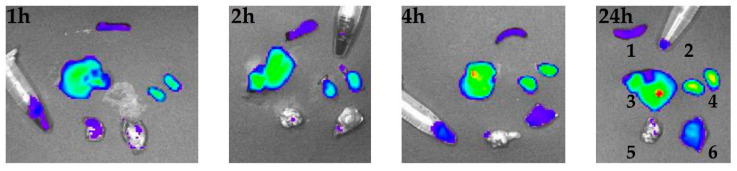
Cy7 fluorescence intensity of the organs and the blood samples of the most representative animal from each time point. Identities: 1—spleen; 2—blood sample in a microtube; 3—liver; 4—kidney; 5—brain; 6—tumour.

**Figure 14 ijms-23-03297-f014:**
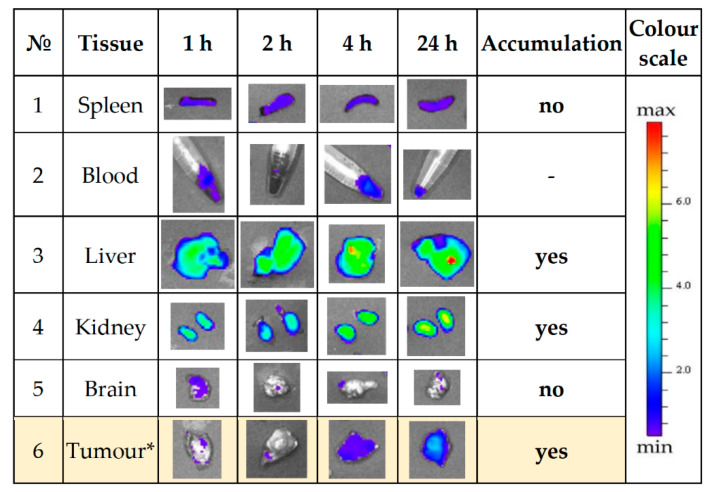
Images illustrating Cy7 fluorescence intensity in the organs and the blood samples of the most representative animal from each time point. * The tumour was excised alongside a piece of surrounding tissue and is visible in the centre of the image as a round sphere.

**Table 1 ijms-23-03297-t001:** The physical and chemical properties of the designed MBS-proteins compared with bovine serum albumin (BSA) (calculated by Protparam software).

	E2-13W4	E1-W8	E1-W12	EmGFP	BSA
MW, kDa	52.2	68.4	84.6	26.9	69.3
Theoretical pI	5.9	6.01	6.03	5.71	5.82
Ext. coefficient at 280 nm (1 g/L)	0.6	0.5	0.5	0.8	0.7
Estimated half-life:					
-Lysate of mammalian reticulocytes, in vitro	30 h	30 h	30 h	30 h	
-Yeast, within the cells	20 h	20 h	>20 h	>20 h	
-Escherichia coli, within the cells	10 h	10 h	>10 h	>10 h	
The instability index	31.07 (stable)	33.55 (stable)	31.07 (stable)	27.73 (stable)	40.28 (unstable)
Aliphatic index	55.85	53.44	52.07	77.82	77.46
Grand average of hydropathicity (GRAVY)	−0.665	−0.679	−0.690	−0.491	−0.429
Estimated charge at pH 7.00	−6.2	−5.2	−5.1	−6.6	−12.2

**Table 2 ijms-23-03297-t002:** Amino acid composition of the functional units of the designed proteins.

Symbol	Function	Source (Amino Acid)	Sequence (Amino Acid)
M	Metal binding site	Calmodulin (human) (D21–L33)	DKDGDGTITTKEL
ELP	Forms the secondary structure	Elastin (human)	[VPGSG]
		Immunoglobulin heavy chain junction region	[VPGYG]
L	Linker	Small antibody fragment linker (single-chain Fv fragment)	[G4S]2[G3S]1, [SG]5GS
F3	Ligand of tumour receptors	Non-histone chromosomal protein HMG-17 (human)	KDEPQRRSARLSAKPA PPKPEPKPKKAPAKK
RGD	Ligand of tumour receptors	Fibronectin (human)	AVTGRGD

**Table 3 ijms-23-03297-t003:** The multiplying capacity of the initial plasmids for each essential motif per cloning cycle.

	Motif				Backbone Vector
Plasmid	MBS	F3	RGD	ELP	
pB7	2	-	-	6	pJET1.2
pFLB77	4	1	-	14	pJET1.2
pRGD1(1)	-	-	1	1	pRSET-EmGFP

Enhancing protein yield—the pE2-13W4 plasmid.

**Table 4 ijms-23-03297-t004:** The number of motifs in each plasmid.

Plasmid	RGD	F3	ELP	MBS	EmGFP
pE2-13W4	3	1	5	4	1
pE1-W8	3	2	10	8	1
pE1-W12	3	3	15	12	1

**Table 5 ijms-23-03297-t005:** Measurement results from the Malvern software.

Sample Name	MW, kDa	“Monomer”, d., nm	% Mass, d., nm
E2-13W4	52.2	9.2	99.87
E1-W8	68.4	10.7	99.98
E1-W12	84.6	11.3	99.99
BSA	69.3	4.7	100.00

**Table 6 ijms-23-03297-t006:** Mean geometric dissociation constants of MBS–Gd^3+^ complexes of proteins in 10 mM HEPES (pH 7.0) at 25 °C.

Protein	mKd (±Standard Deviation), µM
E2-13W4	0.21 ± 0.03
E1-W8	0.17 ± 0.02
E1-W12	0.19 ± 0.04

**Table 7 ijms-23-03297-t007:** Relaxivity values of protein-Gd complexes at pH 7.0.

	E2-13W4-Gd_4_	E1-W8-Gd_8_	E1-W12-Gd_12_	Magnevist^®^
r1, mM^−1^ s^−1^	6.84	6.61	6.66	4.43

**Table 8 ijms-23-03297-t008:** Concentrations of the samples and their functional elements during exposure and the Gd accumulation ratio of tumour to normal cells.

	Functional Elements of the Samples in the Cell Incubation Medium				Results	
Sample Name	[Protein], µM	[RGD] *, µM	[F3] **, µM	[Gd], µM	U87/Fb Ratio	A375/Fb Ratio
E2-13W4-Gd_4_	2.50	7.50	2.50	10	4.4	2.3
E1-W8-Gd_8_	1.25	3.75	2.50	10	4.1	2.6
E1-W12-Gd_12_	0.85	2.55	2.55	10	3.0	2.4
Gd(NO_3_)_3_	10	-	-	10	1.1	1.1
Magnevist^®^	10	-	-	10	1.1	1.0

* determined as [Protein] × (RGD motif number per molecule); see Table 4. ** determined as [Protein] × (F3 peptide number per molecule), Table 4.

## Data Availability

Not applicable.

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
