# Peer review of "Using ELP Repeats as a Scaffold for De Novo Construction of Gadolinium-Binding Domains within Multifunctional Recombinant Proteins for Targeted Delivery of Gadolinium to Tumour Cells"

_ijms, 2022, doi:10.3390/ijms23063297_

Round 1

Reviewer 1 Report

This paper describes the design of three artificial proteins as chelating agent of gadolinium for  radiotherapy and MRI use, and featuring cancer targeting abilities via RGD or F3 peptide ligands. Many information is present in the manuscript which made it not smooth to read. However, overall the work is very interesting and all procedures were fairly described in details.

This referee has few issues to consider

  • How the different protein complexes affect the tumoral and healthy cells cytotoxicity ?
  • In Figure 15, the uptake of the E2-13W4 protein by U87MG Glioma cells is shown in presence and in absence of Mn2+, the authors claim the successful targeting efficiency of RGD ligand, however no control experiment is provided. The authors should compare their results with the uptake of the protein without the targeting ligand to confirm the specificity.
  • Same comment for Figures 18 and 19, showing the biodistribution of Cy7 labeled E1-W12 over time for different groups of animals. A control experiment of the protein without the F3 peptide would strengthen the conclusions.

Reviewer 2 Report

In the manuscript entitled “Using ELP repeats as a scaffold for de novo construction of gadolinium-binding domains within multifunctional recombinant 3 proteins for targeted delivery of gadolinium to tumor cells” Shevelev and co-workers report an approach for designing Gadulinium binding proteins in which they optimized the number of metal-binding sites and tumor ligan peptides. The authors claim that this alternative approach allows overcoming the major limitations related to the recombinant technology for producing contrast agents for BRT/MRI investigation. Overall, this is an interesting work that may provide novel insights into the research for new, safe methodologies for cancer diagnosis and treatment. In my opinion, the authors have to address the following major points before publication:

  • The main limitation of this study is the lack of binding affinity constants between the designed three Gd-binding proteins and the two membrane receptors (αvβ3 integrin and nucleolin). A fundamental requirement for tumor-specific ligands is that they have to show high selectivity as well as high affinity for a specific receptor present on the cell surface of tumors.
  • The authors have to reshape the Results and discussion: there are too many technical details that can be easily moved to the material and methods section. In addition, the figures from 1 to 5 can be included as panels in just one figure. The figures from 8 to 10 can be reported as supplementary information. The insertion of a flow chart describing the complex approach can help the reader otherwise is really hard to go through the procedure used for the optimization of the protocol for designing  Gd-binding proteins.
  • The quality of all figures has to be improved. For example, the insertion of labels in Figures 2 and 5 is necessary.

Round 2

Reviewer 2 Report

The quality of the revised version of the manuscript is significantly improved. The manuscript is suitable for publication.